# Casein Kinase 1α—A Target for Prostate Cancer Therapy?

**DOI:** 10.3390/cancers16132436

**Published:** 2024-07-02

**Authors:** Emma Lishman-Walker, Kelly Coffey

**Affiliations:** Biosciences Institute, Newcastle Cancer Centre, Newcastle University, Newcastle upon Tyne NE2 4HH, UK; emma.lishman-walker@newcastle.ac.uk

**Keywords:** casein kinase, prostate cancer, cell signalling, androgen receptor

## Abstract

**Simple Summary:**

Prostate cancer (PCa) is the most common cancer in males in the UK, resulting in more than 12,000 deaths per year in the UK. A large proportion of patients stop responding to currently available treatments; consequently, identifying alternative treatment options is key to prolonging the life of these men and improving survival rates. Protein kinases regulate signalling pathways, which are often dysregulated in cancer, leading to tumour growth. Additionally, kinase driven phosphorylation regulates androgen receptor activity, which is a key driver of PCa. Understanding how these signalling pathways contribute towards PCa is critical in the development of novel treatment options. The kinase, casein kinase 1 alpha (CK1α), has potential as a PCa therapeutic. Here we describe the current literature supporting this hypothesis and identify where further research is needed to fully understand the functional role and therapeutic potential of CK1α in PCa.

**Abstract:**

The androgen receptor (AR) is a key driver of prostate cancer (PCa) and, as such, current mainstay treatments target this molecule. However, resistance commonly arises to these therapies and, therefore, additional targets must be evaluated to improve patient outcomes. Consequently, alternative approaches for indirectly targeting the AR are sought. AR crosstalk with other signalling pathways, including several protein kinase signalling cascades, has been identified as a potential route to combat therapy resistance. The casein kinase 1 (CK1) family of protein kinases phosphorylate a multitude of substrates, allowing them to regulate a diverse range of pathways from the cell cycle to DNA damage repair. As well as its role in several signalling pathways that are de-regulated in PCa, mutational data suggest its potential to promote prostate carcinogenesis. CK1α is one isoform predicted to regulate AR activity via phosphorylation and has been implicated in the progression of several other cancer types. In this review, we explore how the normal biological function of CK1 is de-regulated in cancer, the impact on signalling pathways and how this contributes towards prostate tumourigenesis, with a particular focus on the CK1α isoform as a novel therapeutic target for PCa.

## 1. Introduction

Prostate cancer (PCa) results in the deaths of more than 12,000 men each year, with as many as one in six men diagnosed throughout their lifetime. Indeed, incidence rates are expected to rise, increasing by 15% by 2040 due to an aging population and improved detection methods [1]. A key driver of PCa is androgen receptor (AR) signalling (Figure 1); therefore, therapies targeting this pathway are central to the management of advanced PCa [2]. In particular, androgen deprivation therapy (ADT) is used to reduce the levels of the AR ligand, testosterone, in the body to switch off the AR signalling pathway. Whilst this may be initially effective, resistance to these treatments can arise through multiple mechanisms, including AR amplification or mutation, AR splice variant production and alterations to AR co-regulators, demonstrating that AR is still a tractable therapeutic target in therapy resistant disease. Patients that are no longer responsive to ADT are described as having castration resistant PCa (CRPC) [3]. Unfortunately, for the 10–20% of patients which progress to CRPC, the outlook is poor despite the development of second-generation anti-androgen treatments such as enzalutamide, which target the AR directly to prevent aberrant signalling. This highlights the unmet clinical need for novel therapies to be used alone or in combination with existing ADT. To achieve this, additional key drivers which are potentially targetable need to be identified and characterised within the context of PCa.

The activity of the AR is partly regulated through post-translational modifications [4]. Deregulation of the proteins responsible for these modifications can result in aberrant AR activity [4]. The most well characterised AR post-translational modification is phosphorylation. Crosstalk between the AR signalling pathway and protein kinase signalling cascades also represents an area for further investigation to identify indirect approaches to targeting aberrant AR signalling in PCa (Figure 1). This strategy could help to overcome treatment resistance in response to current anti-androgen therapies. One such kinase of interest is casein kinase 1 alpha (CK1α). Several publications have highlighted the importance of this kinase in biological functions and the de-regulation of these pathways in cancer and neurodegenerative diseases [5,6,7,8,9]. The aim of this review is to focus upon the role of CK1α in the context of PCa, explore how it can alter critical oncogenic signalling pathways, and provide a rationale for targeting CK1α in PCa.

## 2. CK1 Protein Kinase Family

The CK1 proteins are a family of seven ATP-dependent monomeric serine/threonine kinases (α, β, γ1, γ2, γ3, δ, and ε), which are important in many physiological signal transduction pathways. The kinase domain of the CK1 family is highly conserved across eukaryotes; however, there is some variation within the N and C-terminals (Figure 2A) [10,11]. This structural conservation enables multiple CK1 isoforms to phosphorylate the same substrate. Despite this, there are several examples of distinct and even opposing roles between different CK1 isoforms within the same signalling cascade.

The seven CK1 family isoforms are encoded by distinct genes, and additional splice variants have been reported for some isoforms. Whilst the structure is highly conserved between all isoforms, there are two distinct clusters with the α, δ, and ε isoforms being more closely related to each other than the three γ isoforms (Figure 2B). Expression of all CK1 isoforms, other than CK1β, has been detected at both the mRNA and protein level in most mammalian tissues. The ubiquitous expression of these proteins, in addition to the expansive range of substrates, speaks to the diversity of this kinase family and their importance in a wide range of biological processes, including cell cycle regulation, DNA damage response (DDR), cell differentiation, immune response, apoptosis, and autophagy. Hence, if the activity of CK1 family members is not tightly regulated then key biological processes are de-regulated. This highlights the potential pathogenicity of CK1 kinases, generating interest in the exploitation of CK1 biology as a therapeutic option for cancer and neurodegenerative diseases.

## 3. CK1α Protein Structure

The crystal structure of mammalian CK1α when bound to small molecules has been resolved using X-ray diffraction, which will assist in the design of targeted agents [13,14]. In addition to multiple CK1 isoforms, the CK1α gene, *CSNK1A1*, can be alternatively spliced to produce four variants. The literature surrounding these variants can be difficult to navigate, with some articles describing only the first two variants discovered (CK1α and CK1αL) [15]. The nomenclature also differs between publications, here described by the inclusion of the long (L) or short (S) peptides (CK1α, CK1αS, CK1αL and CK1αLS), which are 28 and 12 amino acids long, respectively (Figure 3) [16]. However, variants have also been described as CK1αNI (no insert, CK1α equivalent), CK1αS, CK1αL and CK1αSN (CK1αL with an N-terminal truncation) [17]. In this review, we will use the L and S nomenclature.

CK1αL variants contain a nuclear localisation signal (NLS), and distinct subcellular localisation of variants has been observed in both zebrafish and mammalian models [20,21]. However, immunofluorescent data from the Human Protein Atlas [22], across three different mammalian cell lines, shows that the majority of CK1α is localised in the cytosol. This implies that the isoforms lacking the NLS are most abundant in human cell lines or the NLS is blocked. Interestingly, when comparing the expression of CK1α and CK1αL cDNA within multiple rat tissues, CK1α cDNA was most abundant [15], providing further evidence that cytosolic forms are the most prevalent. Biochemical properties also differ between CK1α variants, for example casein and phosvitin substrate kinetics vary between the L and S forms [15]. Furthermore, pyrvinium, which activates CK1α through either direct or allosteric binding, elicits a greater level of β-catenin phosphorylation (S45) compared to variants lacking the L peptide (CK1αL and CK1αLS) [23]. Such differences in kinase activity could be exploited to modulate signalling cascades across multiple cell types, dependent upon the prevalence of each variant in differing contexts.

## 4. CK1α Regulation

The CK1 kinases are traditionally described as constitutively active; however, recent studies provide evidence for kinase activity regulatory mechanisms, suggesting that this may not be the case. Regulation by phosphorylation/dephosphorylation, spatial regulation, binding partners and substrate availability have been reported (Figure 4).

### 4.1. Phospho-Regulation

Post-translational modifications can negatively regulate CK1 activity, with autophosphorylation being a critical mechanism. Auto-phosphorylation sites have been identified in the C-terminal region of a subset of CK1 isoforms, including CK1α [24,25]. Indeed, the CK1αL variants have additional auto-phosphorylation sites at S156 and T321 as shown in Figure 4 [26,27,28]. Furthermore, studies in the human and yeast orthologues of CK1δ/ε have revealed a conserved autophosphorylation site which impedes kinase activity upon phosphorylation by altering protein structure and potentially affecting substrate recognition [29]. The high degree of homology across CK1 proteins may suggest that substrate recognition is similarly impaired by phosphorylation across all isoforms.

Other protein kinases, such as protein kinase Cα, regulate the activity of CK1δ through phosphorylation of three possible C-terminal residues [30]. Other kinases identified to phosphorylate CK1δ and subsequently regulate its activity include Checkpoint Kinase 1 and Protein Kinase A [31]. Conversely, dephosphorylation by λ phosphatase in vitro increases CK1α activity and an increased rate of CK1δ/ε kinase activity has been reported following phosphatase treatment [25,29]. In summary, phosphorylation by CK1 itself or external kinases leads to inhibition of CK1 catalytic activity, whereas de-phosphorylation promotes activity.

### 4.2. Spatial Regulation

Given the range of tissues that express CK1, in addition to the numerous substrates (over 100 reported so far), it seems logical that the cellular context is also essential in regulating CK1 activity. Indeed, CK1 localises to numerous different cellular structures, and this can even differ between splice variants of the same gene. For example, CK1αL variants, which harbour a NLS within the L domain, can enter the nucleus, whereas CK1α and CK1αS variants lacking this NLS reside in the cytoplasm. This partitioning of isoforms is suggestive of differing functions for alternative CK1α splice variants due to their ability to interact with different substrates and co-factors discrete to these compartments.

Direct binding of CK1 isoforms by other proteins can influence CK1 kinase activity and influence its cellular localisation. For example, FAM83H can recruit CK1α to nuclear speckles in a subset of colorectal cancer cell lines, with siRNA knockdown of FAM38H reducing the immunostaining of CK1α at this location. Additionally, SON (a DNA and RNA binding protein) was found to co-localise with these proteins and has been suggested to act as a scaffold to anchor CK1α and FAM83H to nuclear speckles. Knockdown or inhibition of CK1α did not impair the RNA processing function of SON, so the exact function of CK1α at nuclear speckles requires further elucidation [32]. Additionally, other FAM83 family members have been found to colocalise with CK1 isoforms α, δ, and ε, with differing interaction affinities. In particular, immunofluorescent studies demonstrated that CK1α and FAM83 proteins showed distinct localisation patterns dependent upon the FAM83 family member induced. For example, FAM83B produced perinuclear localisation of CK1α [33].

### 4.3. Protein Partners

In addition to regulating the location of CK1α within the cell, protein binding partners can regulate CK1α kinase activity. An important example of such a protein is the RNA helicase DDX3. Four CK1 isoforms (α, δ, ε, and γ2) directly interact with, and are activated by, DDX3. This interaction is particularly important in the Wnt pathway which is often dysregulated in PCa [34]. Specifically, when DDX3 is knocked down in the absence of wnt3a, CK1 activity is inhibited to a similar level as when treated with a CK1 inhibitor such as D4476, suggesting that DDX3 is essential for CK1 activity [35,36]. Despite previously being reported to stimulate CK1α, δ and ε isoforms equally [35], a more recent publication suggests that DDX3 exerts a greater regulatory function upon CK1ε than the other two isoforms studied, at least in HEK293T cells [36]. Although it is possible that the regulatory activity varies across different cell types depending upon the predominant CK1 isoform expressed.

The relationship between CK1, DDX3 and Wnt signalling provides a rationale for therapeutic targeting of CK1 in PCa. Exploiting the proteins that recruit or regulate CK1 activity within subcellular locations may present further opportunities for location specific therapeutic targeting of CK1.

### 4.4. Substrate Availability

CK1 kinases preferentially recognise phosphorylated or acidic substrates, the availability of these substrates consequently affects CK1 activity. The impact of substrate availability on kinase activity became evident when modelled in *E. coli*, as CK1 auto-phosphorylation rate varied dependent upon the levels of available substrate [25].

As expected of a Ser/Thr kinase, the canonical consensus recognition motif for CK1 contains a phosphorylated serine (S) or threonine (T) residue as follows: pS/T-X-X-(X)-S/T; however, additional non-canonical motifs are also recognised. This kinase family consists of acidophiles and therefore highly acidic residues can act as substrates, although phosphorylated targets are preferred [6].

In summary, the regulation of CK1 kinases is complex, dependent upon post-translational modifications, cellular location and substrate availability as well as recruitment by other proteins. These mechanisms are likely to work cooperatively rather than exclusively to modulate CK1 activity.

## 5. CK1 and Disease

CK1α plays a role in a range of biological processes within healthy cells, including cell proliferation, apoptosis and regulation of genome stability. In cancer (Figure 5) and other disease states, such as neurodegenerative diseases [8], these processes are affected by CK1α de-regulation (Table 1). Furthermore, mutation of *CSNK1A1* is most commonly associated with myelodysplastic syndromes (5q deletion syndrome) and therefore CK1α has been identified as a promising therapeutic target for these patients [37,38].

CK1α has been described as playing a role in the progression of prostate, breast, ovarian, and colorectal cancers to name a few [5,6,7,43]. For some cancer types, a correlation between gene expression and clinical outcome has been identified [7]. For example, the Human Protein Atlas reports high expression of *CSNK1A1* as prognostically favourable in colorectal cancer and unfavourable in pancreatic cancer [22]. However, Richter et al. reported that high expression correlates with poor survival in colorectal cancer, particularly for higher grade tumours [43], suggesting there is still considerable confusion as to the utility of *CSNK1A1* as a prognostic marker and more work to be done, particularly at the protein level.

The cellular environment is likely to be key in determining the role of CK1 in tumourigenesis; an idea supported by research showing that patients with functional p53 had little correlation between outcome and *CSNK1A1* expression in colon cancer. However, patients with low expression of *CSNK1A1* and who were predicted to have non-functional p53 had significantly poorer survival [44]. As discussed later, there is evidence suggesting that CK1α negatively regulates p53 signalling [45,46]; therefore, in the absence of p53 perhaps CK1α acts as a tumour suppressor, and when the expression of *CSNK1A1* is low, tumour survival is enhanced.

CK1 is a potential therapeutic candidate for relapsed or triple negative breast cancer (TNBC). RNA-seq identified *CSNK1D* as widely overexpressed across the different breast cancer subtypes. Similarly, *CSNK1A1* is overexpressed in luminal and HER2+ subtypes compared to normal tissue. Pathway analysis identified Wnt pathway activation in tumours with increased CK1δ levels, whilst SR-3029 (CK1δ/e specific inhibitor) impaired the growth of patient derived xenografts and decreased expression of Wnt pathway proteins [41]. Additionally, primary tumours that metastasised to lymph nodes showed higher CK1δ expression, whilst knockdown or inhibition of *CSNK1D* reduced wound closure time and invasion of MDA-MB-231 cells, suggesting that CK1δ promotes cancer cell migration [40].

Breast cancer, like PCa, is a hormonally driven cancer and therefore bears a number of biological similarities. The hormonal therapy that is commonly used to treat breast cancer is 4-Hydroxytamoxifen (4-OHT). *CSNK1G2* has been identified to contribute towards 4-OHT sensitivity in oestrogen receptor positive (ER+) cells only. Rescue experiments in ER- cell lines demonstrated that *CSNK1G2* requires ER to affect 4-OHT sensitivity, suggesting some interaction between the hormone receptor and CK1γ2. Oestrogen receptor phosphorylation was reduced in the presence of the CK1 inhibitor D4476; however, it is not clear whether this is a direct result of reduced CK1 activity or indirectly through the PI3K/AKT/mTOR pathway, which was also altered following CK1 inhibition [47]. However, there is evidence in the ER-expressing ovarian cancer Ishikawa cell line that CK1 can contribute to ER phosphorylation as well as epidermal growth factor receptor (EGFR) phosphorylation to regulate their cytosolic localisation and therefore functional outputs including proliferation, migration and sphere forming abilities [48]. This study suggests that the combination of CK1 inhibition with 4-OHT therapy may reduce the risk of endometrial carcinoma for breast cancer patients who are undergoing long-term 4-OHT treatment. Due to the similarities in steroid hormone receptor regulation mechanisms, investigation into whether there are similar mechanisms with AR in PCa is required.

## 6. Why Is CK1α Relevant to PCa?

In other tumour types a number of signalling pathways relevant to PCa development and progression are affected by CK1α activity, suggesting that CK1α may also be an important molecule in PCa too. Indeed, our interrogation of publicly available datasets described below indicates associations with PCa biology supporting this hypothesis.

Transcriptomic data available from the Human Protein Atlas show that *CSNK1A1* is the isoform predominantly expressed in normal prostate tissue (Table 2) [22]. Assessing *CSNK1A1* expression in the Tomlins et al. 2007 dataset [49] revealed that *CSNK1A1* was significantly upregulated in cancer samples compared to PCa precursor samples (adjusted *p*-value of 3.793 × 10^−4^ logFC 0.401), although this dataset is quite small. Additionally, the COSMIC database reports that 2.81% and 0.6% (498 samples) of prostate tumours have upregulated or down-regulation *CSNK1A1* expression, respectively, in the TCGA dataset [50,51]. The genetic alteration frequency of *CSNK1A1* in PCa is lower than that of other therapeutically relevant genes, such as *PRKDC*, and the percentage of structural alterations is similar for metastatic and primary PCa (Figure 6). Hence, it is not clear at which stage of disease CK1α activity may be most relevant and therefore further investigation into this is required.

CK1α does show correlations with AR biology, providing further justification for interrogation as a putative therapeutic target for PCa. Specifically, within the TCGA cohort a positive correlation between *CSNK1A1* and AR expression is observed (Figure 7) [55]. Interestingly, CK1 is predicted to phosphorylate the AR at several serine or threonine sites (Table 3) [56,57], but experimental evidence is lacking to support this, including interaction studies between AR and CK1α. Interestingly, a recent publication identified *CSNK1A1* as the top hit in a CRISPR-Cas9 kinome-wide screen which enabled re-sensitisation to the anti-androgen, enzalutamide, in a castration-resistant PCa (CRPC) cell line CWR22rv1. This highlights the promise of targeting kinases such as CK1α in addition to currently available treatments [39].

Despite a low frequency of genetic aberrations and a small proportion of tumours showing transcriptional alterations, *CSNK1A1* may still be an important driver of PCa for a subset of patients. Furthermore, not only are gene expression studies important, but protein level expression of CK1α must be studied within PCa to truly understand the potential of therapeutically targeting this kinase.

## 7. CK1α Function and Dysregulation in PCa

CK1 kinases either directly activate or inhibit their substrates through phosphorylation or act as upstream regulators to prime other kinases to exert their function. To understand the role of CK1α in PCa we must first consider the normal biological functions of the CK1 kinases. These roles are extensive and varied and within this review only a subset of CK1-affected signalling pathways will be described which have a particular relevance to PCa. More extensive reviews of CK1 kinase functions are available elsewhere [11,42].

### 7.1. Wnt Signalling

Under normal circumstances Wnt signalling is tightly regulated to enable normal cell growth and differentiation. CK1α is an important negative regulator molecule in canonical Wnt signalling, along with axin and glycogen synthase 3B (GSK3B). In the absence of Wnt, CK1α phosphorylates β-catenin at S45, priming it for further phosphorylation by GSK3B at residues T41, S37, and S33. Subsequent degradation via the proteasome maintains low cytoplasmic levels of β-catenin (Figure 8). β-catenin is an example of a non-phosphorylation primed CK1α substrate. Instead, it contains an acidic cluster 5–11 residues downstream of S45 and a hydrophobic side chain in the +1 position. Furthermore, Y145 within the first Armadillo domain of the protein is important for successful phosphorylation at S45 as a consequence of α-catenin binding regulation (Figure 8) [59].

Furthermore, CK1α can phosphorylate axin within the N-terminal tankyrase-binding region which results in an inability of axin to bind tankyrase and thereby become stabilised, thus further promoting the destruction of β-catenin. Four residues, namely S51, S54, S57 and T60, were identified as critical for axin phosphorylation by CK1α and match the consensus CK1 phosphorylation sequence, with T60 being critical for attenuation of axin–tankyrase binding (Figure 8) [60]. Other CK1 isoforms, namely CK1ɛ, CK1δ and CK1γ, have both positive and negative regulatory roles in this signalling cascade, phosphorylating targets such as Dishevelled (Dvl) and Adenomatous Polyposis Coli (APC) [61].

The Wnt signalling cascade is frequently de-regulated in PCa and contributes to tumour growth. Wnt3a elevates β-catenin levels in PCa which, in turn, leads to increased transcription of Wnt target genes and subsequent proliferation. Moreover, independent of androgenic stimulation, Wnt3a conditioned medium promotes growth of LNCaP cells, an androgen dependent PCa cell line model [62]. Furthermore, crosstalk between the AR and β-catenin is known to contribute towards PCa progression and indeed CRPC models display increased levels of AR and β-catenin. Additionally, microarray data show that β-catenin activator genes are upregulated in castrate resistant cells, with β-catenin inhibitor genes being downregulated. However, downstream transcription factors such as Tcf are not increased, implying that β-catenin accumulation is exerting effects through AR activation rather than direct Wnt signalling [63].

As CK1α regulates cytoplasmic β-catenin levels in the absence of Wnt, it may be beneficial to activate CK1α in PCa. However, microarray data suggest that *CSNK1A1* is upregulated in PCa, yet β-catenin expression is also increased, suggesting that elevated CK1α does not impact β-catenin. However, it is possible that post-translational modification results in decreased CK1α protein levels, subsequently allowing β-catenin accumulation. Hence, further work interrogating CK1α protein expression and the impact of CK1α modulation on the Wnt cascade in PCa is required to fully understand the utility of CK1α targeting in this context.

### 7.2. Hippo Signalling

Hippo signalling regulates cell proliferation and cell death, particularly during development, to constrain organ size. Key components of this pathway include YAP and its paralogue TAZ, which enter the nucleus in response to low cell density or lack of cell–cell contact, promoting gene transcription to inhibit apoptosis and promote proliferation when this pathway is turned off. LATS kinase, which phosphorylates YAP/TAZ to turn off transcriptional activity of their associated transcription factors when the Hippo pathway is active, is predicted to prime YAP for phosphorylation by CK1. Indeed, both D4476, a generic CK1 inhibitor, and IC261 (CK1δ/ɛ specific) reduced phospho-YAP S381 levels, and co-expression of active CK1ɛ alongside YAP increased pYAP (S381) compared to kinase-dead CK1ɛ, providing more evidence for the role of CK1ɛ in YAP phosphorylation [64]. Furthermore, CK1δ/ɛ-mediated YAP phosphorylation promotes YAP degradation following ubiquitination by β-TRCP to impair proliferation and migration (Figure 9).

YAP activity has been linked to the invasion and metastasis of melanoma, breast and prostate tumours [65]. Structural changes in the CK1 genes are present in a proportion of metastatic prostate tumours (Figure 9). Therefore, if these changes result in CK1 regulatory dysfunction, the ability of CK1 to negatively regulate YAP activity will be impaired, leading to increased downstream transcriptional activity and contributing to increased migration of prostate tumour cells. Furthermore, bone is a favoured site for PCa metastases, and it is well known that the surface upon which cells grow contributes to the activity of the Hippo signalling pathway and downstream phenotypic characteristics. Indeed, PC3 cells, which are derived from prostatic bone metastases, have altered YAP signalling and cell growth depending on whether a hard or soft growth surface is used [66,67]. However, the role of CK1 in these processes has not been investigated.

In glioma cell lines, CK1α is proposed to modulate migration, with CK1α overexpression promoting wound closure and CK1α inhibition impairing the process. Further investigation revealed increased AKT, pAKT and MMP2 expression in cells overexpressing CK1α as compared to those with reduced CK1α levels. In vivo imaging of mice harbouring tumours derived from cell lines overexpressing CK1α confirmed increased tumour size and migration [68]. Indirect regulation of AKT by CK1α has been reported previously, with DEPTOR being phosphorylated by CK1α, resulting in the activation of mTOR signalling [30,69]. Interaction between YAP and AKT has also been reported [70], hence further investigation to untangle the interactions between CK1α, AKT and YAP is required. However, if CK1α regulates AKT activity, which subsequently leads to YAP degradation, targeting CK1α may prove beneficial in tumours with overactive YAP signalling.

Interestingly there appears to be crosstalk between the Wnt signalling cascade and YAP. Wnt3a increases expression and nuclear translocation of AR and YAP in LNCaP cells to promote cell growth in androgen depleted conditions [71]. Given that CK1α acts downstream of Wnt3a to inhibit Wnt signalling, the activation of CK1α is unlikely to also promote phosphorylation and degradation of YAP. CK1δ/ɛ, however, are thought to directly phosphorylate YAP. At present CK1δ/ɛ activators remain unavailable; perhaps development of additional CK1 activators could enable both the inhibition of Wnt signalling and Hippo signalling in PCa.

### 7.3. Cell Cycle Regulation

The role of CK1 in cell cycle regulation has recently been reviewed [72]. Like many cancers, cell cycle de-regulation contributes to prostate tumourigenesis [72]. Other cell cycle regulators such as AURKA and PLK1 are therapeutic targets, with several ongoing clinical trials incorporating the use of agents targeting these kinases [73,74,75,76].

Whilst CK1α demonstrates consistent expression levels throughout the cell cycle, it does display cell cycle dependent patterns of subcellular localisation, suggesting a role in spindle positioning. For example, the FAM83 protein FAM83D directly interacts with and recruits CK1α to the mitotic spindle, but is not observed in asynchronous U20S, HeLa or A549 cell lines [33]. Furthermore, mutations in the CK1α interacting DUF1699 domain of FAM83D impairs CK1α co-localisation with FAM83 and with the spindle itself. Combined mass spectrometry, immunoblotting and immunofluorescence microscopy data imply that CK1α is involved in regulating the mitotic spindle and that CK1α requires additional proteins to regulate the cell cycle [33,77].

Similarly, other CK1 isoforms have been implicated at the mitotic spindle. FAM110A, another spindle pole associated protein involved in chromosomal alignment, has been shown to interact with, and be phosphorylated by CK1δ/ɛ. However, depletion of *CSNK1E* had no impact upon FAM110A function, unlike *CSNK1D*, suggesting functional redundancy of CK1ɛ. Furthermore, pharmacological inhibition of CK1 impaired mitotic progression [78]. CK1δ has also been associated with the centrosome, kinetochore and microtubules; however, the exact contribution of each CK1 isoform in cell division requires further elucidation.

The anaphase promoting complex (APC) can be regulated by Chd1 or CDC20, which are considered oncogenic proteins in PCa. CK1δ contains functional RXXL motifs, which APC^Cdh1^ uses to recognise substrates, resulting in ubiquitination and turnover of CK1δ in cerebellar granule cell progenitors [79]. The authors did not assess the role of CDC20 in regulating CK1 in their model; however, CDC20 can interact with Rb1 [80] and CK1ε phosphorylates Rb1. Therefore, there may be some interaction between CDC20 and CK1ε via Rb1. Given the evidence for APC^Chd1^ in regulating CK1δ turnover, it would be interesting to see if CDC20 has a similar role in PCa.

### 7.4. DNA Damage Response

The tumour suppressor gene, *TP53*, plays an essential role in cell cycle regulation, apoptosis, and the DNA damage response (DDR). Cellular stress, such as DNA damage, activates p53 signalling to maintain genome stability. Several CK1 isoforms can regulate the activity of p53 signalling cascade proteins including MDM2, MDMX and p53 itself through an activating phosphorylation at S9 (Figure 10) [81]. However, the outcome of these interactions is dictated by the interacting CK1 isoform. For example, MDMX, expressed in the absence of CK1α, increases p53 activity, suggesting that CK1α plays a role in MDMX-mediated p53 inhibition, contrary to the reported p53 activation by CK1δ/ε. Furthermore, kinase-dead CK1α mutants were unable to stimulate MDMX activity, and CK1 inhibition greatly reduced MDMX phosphorylation in MCF7 cells. This provides evidence for a functional role for CK1α-mediated MDMX phosphorylation with S289 being identified as particularly important [45]. Indeed, p53 can also be phosphorylated by multiple CK1 isoforms in primed phosphorylation and non-primed phosphorylation events (Figure 10) [82].

Multiple studies have shown that siRNA knockdown of CK1α increases p53, p21, and MDM2 expression, whilst E2F1 and pRb1 expression is decreased [46]. Whilst the changes to p21 and MDM2 are thought to be p53 mediated, E2F1 appears to be regulated independently of p53, as in p53 null cells E2F1 continued to decrease in response to D4476 treatment [46]. Loss of Rb1, a transcriptional co-repressor involved in cell cycle regulation, is associated with poorer outcomes in several cancers, including PCa [83]. These off-target effects of inhibiting CK1 must be considered when determining whether this is a suitable therapeutic approach.

Like many cancers, defective components of DDR pathways have been identified in PCa, making it an attractive therapeutic target. Impaired DDR is associated with more aggressive PCa, and the identification of crosstalk between AR and DNA-PKcs has ultimately resulted in an ongoing clinical trial assessing the benefit of combination treatment with an experimental DNA-PK inhibitor (CC-115) and enzalutamide (NCT02833883) [84]. Therapy resistance is a growing concern in PCa research, and as such the mechanisms of resistance to key therapeutics such as enzalutamide are under investigation. CK1α contributes towards enzalutamide resistance in a cell line model of CRPC and, through gene-set enrichment analysis (GSEA), DDR pathways were found to be significantly downregulated when *CSNK1A1* was overexpressed. Expression of ATM, a key molecule in double strand break repair, was shown to inversely correlate with CK1α in patient samples using immunohistochemistry. Furthermore, compared to wild-type ATM, mutated ATM showed reduced CK1α-mediated phosphorylation and CK1α inhibition reduced ATM phosphorylation [39], suggesting that ATM is a substrate for CK1α. For PCa patients overexpressing CK1α, this heightened level of p-ATM results in enhanced ATM turnover, which contributes to therapy resistance. Use of CK1α inhibitors in the clinic may provide patient benefit by delaying or even reversing enzalutamide resistance. However, as with all targeted therapies, the efficacy of such treatment will be dependent upon the molecular characteristics of each patient’s tumour. For example, the authors show that whilst knockdown of *CSNK1A1* promotes sensitivity to enzalutamide, additional knockdown of *TP53* can alleviate this effect somewhat, suggesting that for some patients CK1α inhibition may be less effective.

### 7.5. Autophagy and Cell Death

Autophagy is key in maintaining cellular homeostasis through recycling damaged proteins and organelles. Failure of this pathway ultimately results in cell death. Genes which modulate this process are regulated by the Forkhead Box O (FOXO) transcription factors which are predominantly active in the nucleus. Due to their role in promoting autophagy, they are often considered to be tumour suppressors. Translocation of FOXO proteins to the cytoplasm acts as a repressive mechanism that can be triggered by post-translational modifications [85], including phosphorylation mediated by CK1α.

Whilst FOXO3a is the most highly expressed member of the FOXO family in PCa cell lines, its expression is reduced in the androgen-independent LNCaP (LNCaP-AI) cell line model compared to its parental androgen-dependent LNCaP cell line. Furthermore, in LNCaP-AI cells, the level of pFOXO3a (Thr32) is increased and associated with reduced transcriptional activity in this androgen-independent model [86]. CK1α can phosphorylate FOXO3a at S318/321, resulting in nuclear export and subsequent proteasomal degradation. Indeed, inhibition of CK1 with D4476 reduced the phosphorylation of FOXO1 at S322/325 and subsequently increased nuclear FOXO levels [87] to promote transcription of autophagy-related genes. Similarly, phosphorylation of FOXO1a at S322/325, following a priming phosphorylation event at S319 by AKT, leads to nuclear exclusion [88]. Prostate cell lines LNCaP and PC3 which have higher levels of p-AKT (Ser473) also display increased cytoplasmic FOXO3a and 14-3-3, which binds and sequesters FOXO to the cytoplasm [89]. Furthermore, in PCa, PTEN deletion, among other factors, results in AKT activation, causing increased FOXO nuclear exclusion. Additionally, FOXO1a and FOXO3a levels are increased in transgenic adenocarcinoma of mouse prostate (TRAMP) mice as compared to non-transgenic mice, but the proportion of nuclear FOXO is decreased as compared to non-transgenic mice [90]. Together, this suggests that FOXO activity, and therefore autophagy, is suppressed in more advanced PCa.

In the presence of oncogenic RAS, the PI3K/AKT/mTOR pathway is up-regulated, resulting in increased CK1α activity, which, in turn, reduces nuclear FOXO3a and decreases autophagy [91]. Additionally, nuclear FOXO4 increases with CK1α siRNA-mediated depletion or CK1 kinase inhibition in HCT-116 and SW480 colon cancer cell lines. Here, mutant RAS drives proteasome activity to further deplete FOXO levels, which can be reversed using Bortezomib, an FDA approved 26 S proteasome inhibitor. Combination of Bortezomib and CK1 inhibition synergistically increased FOXO levels in the studied RAS mutant cell lines [92], further demonstrating the negative role of CK1 in FOXO regulation. Whilst mutations in RAS genes are relatively low in PCa [93], ranging from 2.25 to 9.68% of cases across various datasets on cBioPortal (Figure 11), it is thought that RAS de-regulation contributes towards metastasis and, in combination with PTEN loss, can accelerate cancer progression [94]. Given these findings in a proportion of PCas that have both PTEN loss and RAS activation, CK1α activity may be repressing FOXO activity. In such cases, CK1α inhibition could be a potential strategy to reactivate these tumour suppressors and regulate proliferation and autophagy.

More recently, CK1δ and ε have been shown to play an important role in ULK1-mediated autophagy in lung cancer and melanoma cells by regulation of ULK1 levels [95]. It remains to be seen whether these findings are also applicable to PCa biology. Furthermore, CK1α, δ and ε have been found to interact with RIPK3 to result in phosphorylation at S277, resulting in critical recruitment of MLKL upon stimulation of the necroptosis cell death pathway [96]. This phosphorylation is dependent on E221, E225, and RIPK3 oligomerisation (Figure 12) [96].

### 7.6. Inflammation and Immune Response

NF-κB is a transcription factor that promotes inflammation, immune activation, cell growth, proliferation, survival, and apoptosis. Consequently, this pathway has received attention due to its role in tumourigenesis, including PCa, when dysregulated [97,98]. The NF-κB family of proteins are regulated by the inhibitor of NF-κB kinase (IKK) complex proteins that bind to the NLS of NF-κBs, sequestering them in the cytoplasm. CK1α participates in NF-κB regulation by association with the CARMA1/CARD11-BCL10-MALT1 (CBM) complex to subsequently phosphorylate CARD11 (CARMA1), promoting activation and termination of the signalling cascade (Figure 13). Moreover, CARD11 binding was impaired by expression of kinase-dead CK1α. In CK1α knockout cells, downstream signalling was reduced, despite expressing a variant of CARD11 that enables constitutive assembly of the CBM complex. Another component of the CBM complex is MALT1, which, when stimulated, is phosphorylated at S562 and S649, sites predicted to be CK1α substrates. Together, these findings suggest that the kinase activity, rather than just the presence of CK1α, contributes to CBM complex assembly and subsequent signalling [99].

Steroid receptors, including AR, can inhibit NF-κB and inflammatory markers are increased in PCa, suggesting that crosstalk between AR and the IKK proteins may contribute to altered NF-κB signalling in PCa [98]. Similarly, CK1α may play a similar role given that it activates NF-κB signalling and is upregulated in some prostate tumours.

In human T cells, siRNA-mediated knockdown of CK1α reduces IL-2 production, consistent with decreased activation of an NF-κB luciferase reporter, and introduction of kinase-dead mutant CK1α increased T-cell receptor activity; evidence that CK1α can negatively regulate NF-κB signalling to impact the immune response [100].

### 7.7. Regulation of miRNA-Mediated Gene Silencing

MicroRNAs play a significant role in the regulation of cellular signalling pathways, with a number showing altered expression and oncogenic properties in PCa. These short, non-coding RNAs carry out their function by associating with Argonaute (AGO) proteins to form the miRNA-induced silencing complex (miRISC) and target mRNAs with the appropriate complementary sequences for silencing. CK1α can phosphorylate AGO2 within a highly conserved cluster of residues with the C-terminus to prime AGO2 for phosphorylation by other kinases including CK2. This results in hyperphosphorylated AGO2 to regulate association of target mRNAs with miRISC and influence resultant gene silencing. Currently, it is unknown what stimulates AGO2 phosphorylation by CK1α [101].

### 7.8. Testosterone Synthesis

Modulating the levels of testosterone is a key therapeutic intervention for PCa. Conditional knockout of CK1α in mice Leydig cells results in a reduction in testosterone production by reducing levels of CYP17A1, the molecular target of abiraterone, as well as other molecules involved in the testosterone production pathway [102]. Therefore, targeting CK1α in combination with abiraterone could increase its efficacy and allow lower doses to be used.

## 8. Targeting CK1

Due to the increasing interest in CK1 because of its role in pathogenesis, several small-molecule CK1 inhibitors have been developed, some of which are shown in Table 4 (more extensive reviews of CK1 inhibitors are available elsewhere [6,103]). The high degree of homology between the CK1 isoforms poses difficulty in the design and development of drugs to target specific isoforms. Many of the isoform-“specific” drugs target CK1δ/ε, which are the most well-described isoforms. The development of more targeted agents will be key to clinical success given the conflicting roles of CK1 isoforms, such as within the Wnt pathway. Conversely, use of non-specific inhibitors could enable multiple pathways to be modulated concurrently.

The interaction of CK1α with co-factors or scaffolding proteins provides an opportunity for more precise targeting of CK1 than is currently possible with small-molecule inhibitors. Indeed, the interaction between DDX3X and CK1δ/ε has been disrupted using a peptide to alter the Wnt pathway [36]. Similarly, blocking the interaction between MDM2 and CK1α using a peptide resulted in reduced cell growth and viability [108]. The crystal structures of several CK1 isoforms are available to facilitate the design of disrupting peptides, making the use of such agents a possibility in future cancer therapies. These results show the potential of biological agents, both as tools to understand protein–protein interactions, and for their use in future treatment strategies [18,19].

The thalidomide derivative lenalidomide recruits substrates and causes their degradation via the E3 ligase substrate adaptor cereblon. This is achieved by lenalidomide’s association with an exposed β-hairpin motif in its substrates. CK1α contains such a β-hairpin motif, which contains a glycine that is required for cereblon recruitment. However, CK1α is not the only protein with this characteristic, and therefore dual degraders have been developed which target CK1α and Helios (IKZF2), which have demonstrated efficacy in cell lines representing ovarian cancer and diffuse large B cell lymphoma [109,110].

Cancer cells can adapt to develop treatment resistance; one way of addressing this is through combination therapy. Evaluation of CDK4/6 inhibitor resistance in breast cancer cells revealed that post-translational modification leading to loss of *Rb1* was the cause. Inhibition of CK1ε with D4476 or shRNA increased Rb1, which contains predicted CK1 phosphorylation sites. Indeed, mutation of these sites impaired CK1-mediated phosphorylation of Rb1 and reduced the interaction with βTrCP1. CDK4/6 inhibition also upregulates CDK6; however, CK1ε inhibition was able to stop CDK6 accumulation as well as Rb1 depletion [80]. Similarly, CK1α inhibition enhanced the effects of lenalidomide and bortezomib in multiple myeloma cells, resulting in increased apoptosis when compared to single agents, and, importantly, healthy B cells were unaffected [69].

CK1α inhibition can result in decreased MDM2 and increased p53 levels in multiple myeloma cell lines. This results in the degradation of AKT and β-catenin, mediated by elevated p53, thereby decreasing multiple myeloma cell survival [69]. However, the study did not include evaluation of these therapies in p53-null backgrounds which would add further evidence to support the authors’ hypothesis. Interplay between CK1α and p53 means that the effect of CK1α inhibition can be dependent upon p53 status [44]. Moreover, in 2020, a phase I clinical trial (NCT04243785) was launched to investigate the efficacy of BTX-A51, a multi-kinase inhibitor that targets CK1α and CDK7/9 in acute myeloid leukaemia [107]. BTX-A51 is proposed to stabilise p53 and consequently downregulate *MYC*; therefore, the trial was subsequently expanded to include *MYC*-dependent solid tumours. Together, these studies show that CK1α inhibition can improve the efficacy of existing agents, whilst the promise of them as standalone therapies is under investigation [107,111]. However, like many targeted treatments, the genetic and molecular background of tumours will be important in determining the potential of CK1α inhibition for patients.

CK1α activators have also been reported, such as Pyrvinium and SSTC3, which have been discussed above in the context of Wnt signalling [112]. β-catenin levels are frequently high in PCa, suggesting that perhaps activators could provide clinical benefit in modulating Wnt signalling. However, transcriptomic data suggest that *CSNK1A1* is in fact upregulated in PCa. This implies that CK1α may be carefully regulated in PCa to ensure that overexpression does not exceed a level that will be beneficial to the cell. To date, no work has investigated the impact of CK1α activators in PCa models; however, Liu et al. did observe that overexpressing *CSNK1A1* suppressed ATM, contributing to drug resistance [39]. The use of isoform-specific activators may prove an effective tool to unpick the mechanisms by which CK1 activity modulates de-regulated pathways within PCa.

## 9. Conclusions and Future Directions

CK1 is a ubiquitously expressed kinase involved in regulating a wide range of pathways, which, if deregulated, contributes towards oncogenesis. Cell line models have shown the value in targeting CK1 in cancer therapy, such as impaired cell proliferation and migration. Difficulty in targeting CK1 will likely arise from the presence of multiple CK1 isoforms and splice variants, particularly as different isoforms can play opposing roles within the same signalling cascade. Redundancy between isoforms, namely CK1δ/e, must also be considered when selecting the appropriate inhibitor. Many of the pathways CK1 participates in are deregulated in PCa; moreover, transcriptomic data suggest that *CSNK1A1* is more highly expressed in PCa as compared to precursor conditions such as benign prostatic hyperplasia. However, many challenges remain before CK1α interventions can be implemented safely and successfully in the treatment of PCa. There is still a lack of experimental evidence at the protein level, particularly within human tumour specimens, to support the targeting of CK1 isoforms in PCa. Additionally, many studies have been performed in cell lines; however, the impact of targeting a ubiquitously expressed protein involved in many cellular processes must be understood at the organism level. The ongoing clinical trial for BTX-A51 may provide some answers regarding the safety of targeting CK1. Fundamentally, whilst CK1 is linked to several pathways deregulated in PCa, there is no evidence of a direct interaction between AR and CK1 to date. Understanding whether CK1 can modulate AR activity will be important therapeutically; for example, when considering combination treatments and at what stage of tumour progression this will be most effective.

## Figures and Tables

**Figure 1 cancers-16-02436-f001:**
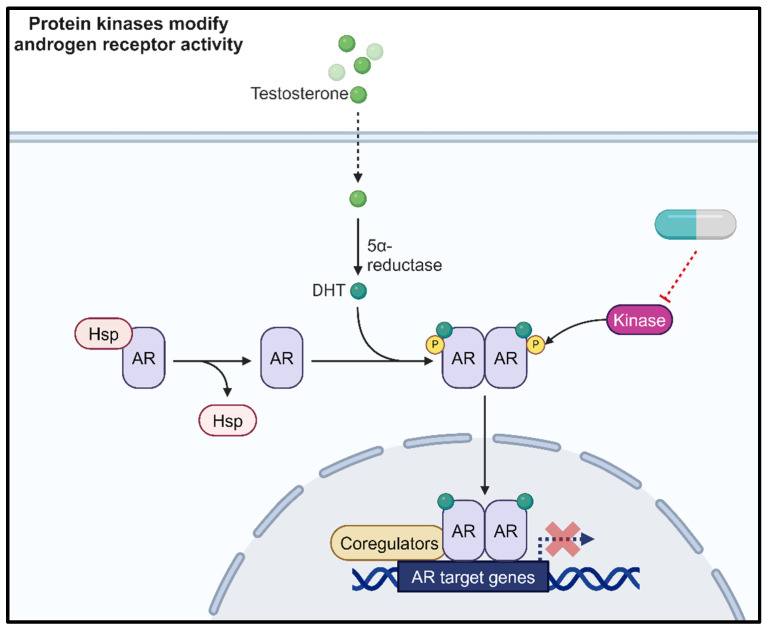
Protein kinases phosphorylate the androgen receptor (AR) to modulate AR activity. Therefore, kinase inhibition represents a novel therapeutic approach. Created with BioRender.com.

**Figure 2 cancers-16-02436-f002:**
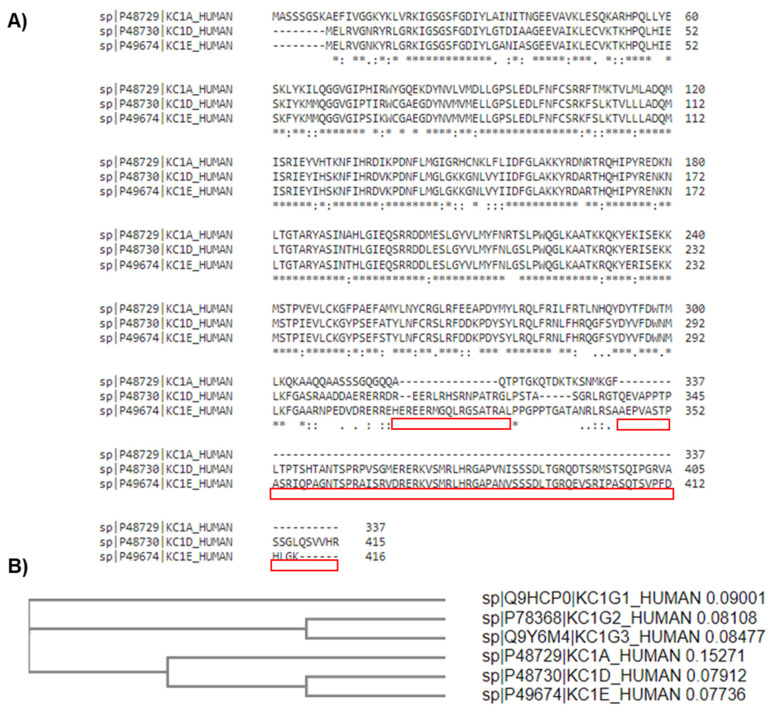
Sequence alignment of CK1 protein isoforms α, δ and ε. (**A**) Highlighted in red are the C-terminal regions where the CK1α sequence diverges from the δ and ε isoforms. (**B**)The phylogenetic tree shows that all six isoforms can be categorised into two clusters: the γ1-3 isoforms show the greatest similarity with each other and α, δ, and ε form a separate cluster. * denotes conserved amino acids. Alignment performed using the Clustal Omega algorithm [12].

**Figure 3 cancers-16-02436-f003:**
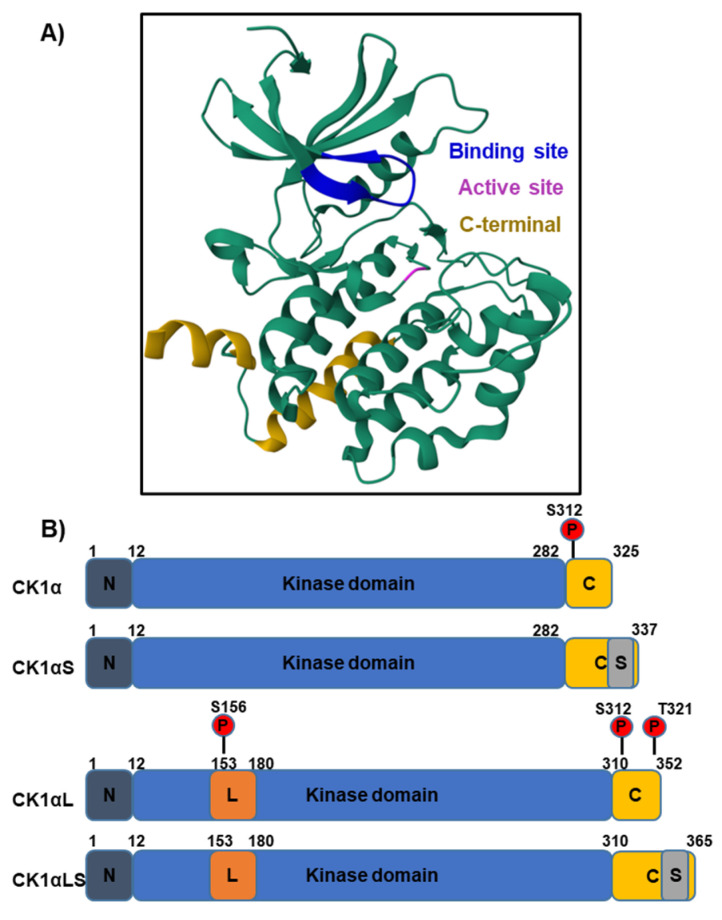
The structure of CK1α bound to the CK1 inhibitor A86, determined by X-ray diffraction [18,19]. (**A**) The crystal secondary structure of CK1α bound to inhibitor A86 is illustrated and the binding site is in dark blue. The magenta-coloured residue is the kinase active site. (**B**) The different CK1α splice variants may include L and S peptide insertions (orange and grey, respectively) and vary in molecular weight from 32 to 42 kDa. The largest isoform contains both inserts and is 365 amino acids long. The L peptide contains a nuclear localisation signal (NLS) to enable translocation into the nucleus. In humans, increased cytosolic CK1α protein levels have been detected, suggesting the isoforms lacking the L-peptide are predominantly expressed. Auto-phosphorylation at S312 serves to regulate kinase activity. The additional auto-phosphorylation sites S156 and T321 (shown in red) are present in variants containing the L-peptide.

**Figure 4 cancers-16-02436-f004:**
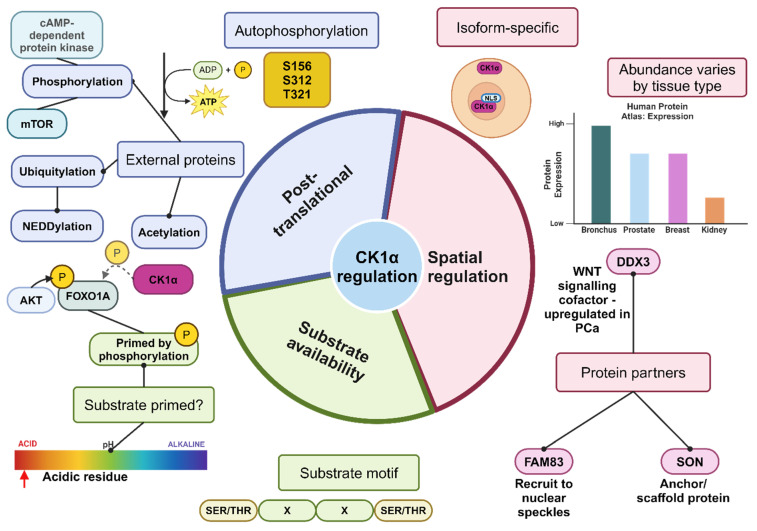
Summary of the mechanisms of regulation of CK1α activity. It is likely that these mechanisms work together to regulate the activity of CK1α. Substrate abundance, as well as whether the substrate has been phosphorylated and primed or contains the recognition motif for CK1α, can impact how readily CK1α functions to phosphorylate the target protein. CK1α is described as ubiquitously expressed; however, there may be differences in subcellular localisation dependent upon the splice variant and whether a NLS is present. Furthermore, the level of protein expression varies by tissue type, and the available co-factors or scaffolding proteins to stabilise and recruit CK1α will vary by tissue type, subsequently affecting the activity of CK1α. Post-translational modification, such as phosphorylation by external proteins, inhibits CK1α activity, whereas dephosphorylation activates CK1α. Created with BioRender.com.

**Figure 5 cancers-16-02436-f005:**
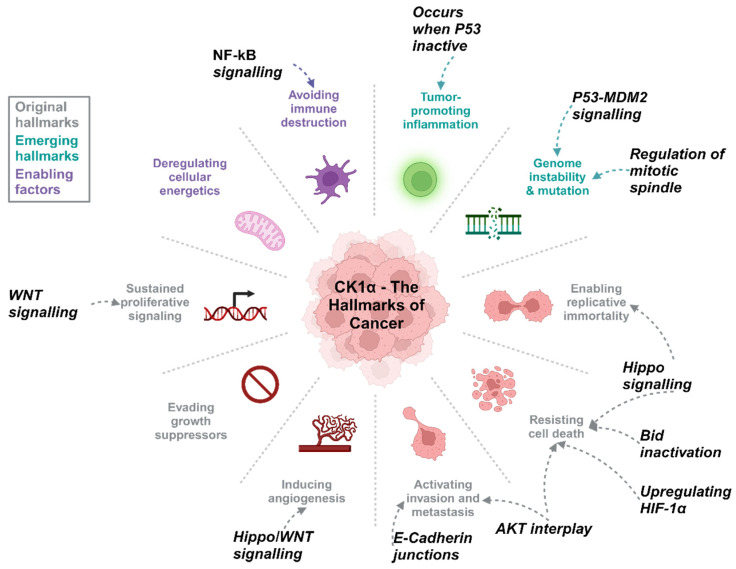
CK1α activity can be associated with the hallmarks of cancer. Wnt and Hippo signalling cascades are frequently de-regulated in PCa and are linked to biological processes such as proliferation and regulation of cell death, CK1α in various other signalling pathways can also promote tumourigenesis. Created with BioRender.com.

**Figure 6 cancers-16-02436-f006:**
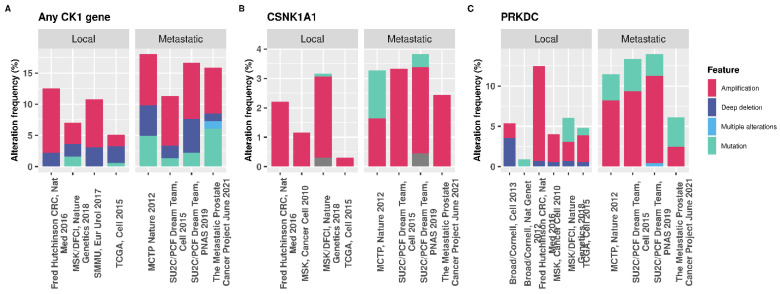
cBioPortal data showing the genetic alteration frequencies in PCa. (**A**) Percentage of patients with a genetic alteration in any CK1 isoform gene (α, δ, ε, γ1, γ2, or γ3). (**B**) *CSNK1A1* only, for which 64/32108 of patients tested harboured a mutation in the gene. (**C**) The percentage of genetic alterations in *PRKDC* in PCa. Out of 32108 patients queried, 190 have a mutation in the gene encoding DNA-PKcs [52,53,54]. The percentage of patients with a mutation in the *PRKDC* gene is greater than for *CSNK1A1*, suggesting that alterations in kinases other than CK1 are more common in PCa which may impact the therapeutic capacity of targeting CK1.

**Figure 7 cancers-16-02436-f007:**
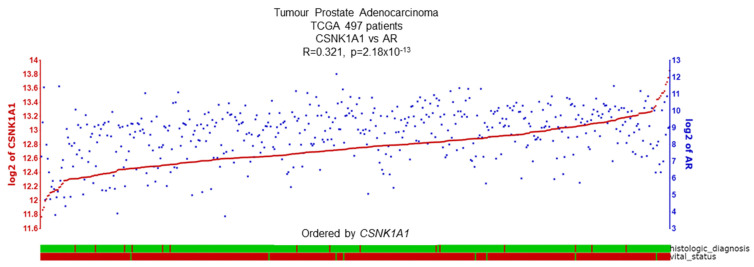
CSNK1A1 and AR expression demonstrate a positive correlation. R2 database shows a significant correlation (R = 0.321, *p* = 2.18 × 10^−13^) between CSNK1A1 and AR expression in the TCGA PCa dataset, including 497 PCa patient samples. Histological status: green = adenocarcinoma, red = another subtype. Vital status: green = deceased, red = alive [58].

**Figure 8 cancers-16-02436-f008:**
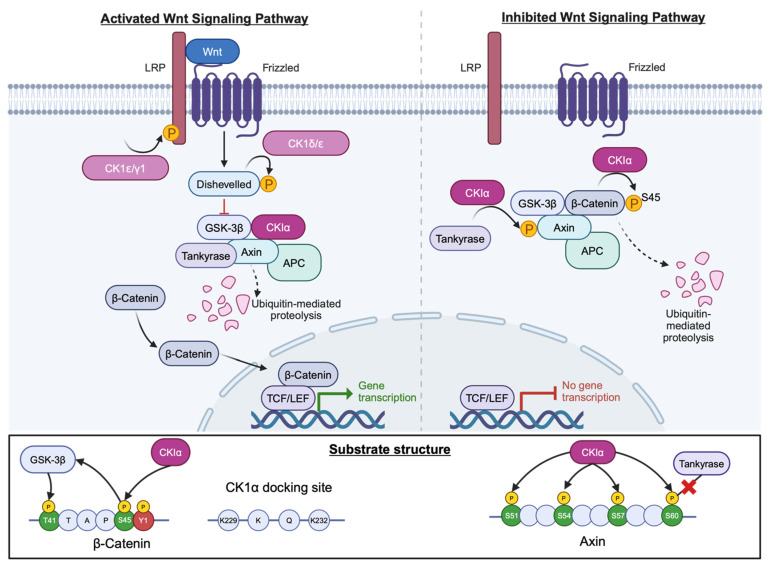
CK1 in the Wnt signalling pathway. Several CK1 isoforms act within this pathway with opposing roles. CK1α (magenta) promotes degradation of β-catenin (pale blue) by phosphorylating at S45 in the absence of Wnt. Created with BioRender.com.

**Figure 9 cancers-16-02436-f009:**
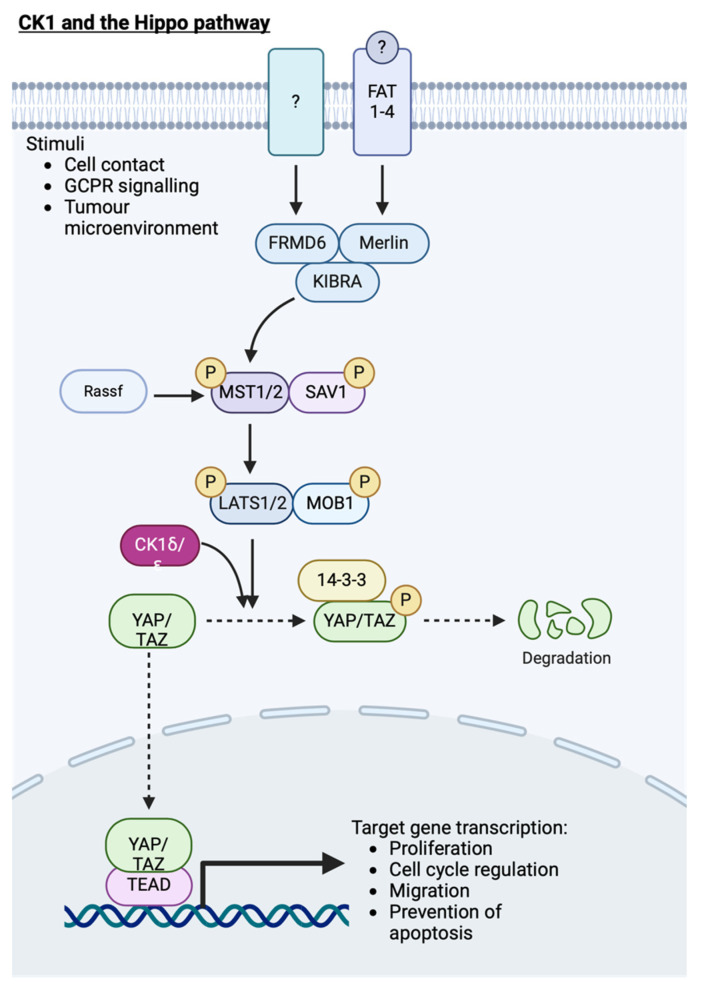
CK1 and its role in the Hippo signalling cascade. Activation of YAP target gene transcription results in proliferation, cell migration and inhibition of apoptosis. Following priming phosphorylation of YAP by LATS1/2, CK1 further phosphorylates YAP, targeting it for degradation. Created with BioRender.com.

**Figure 10 cancers-16-02436-f010:**
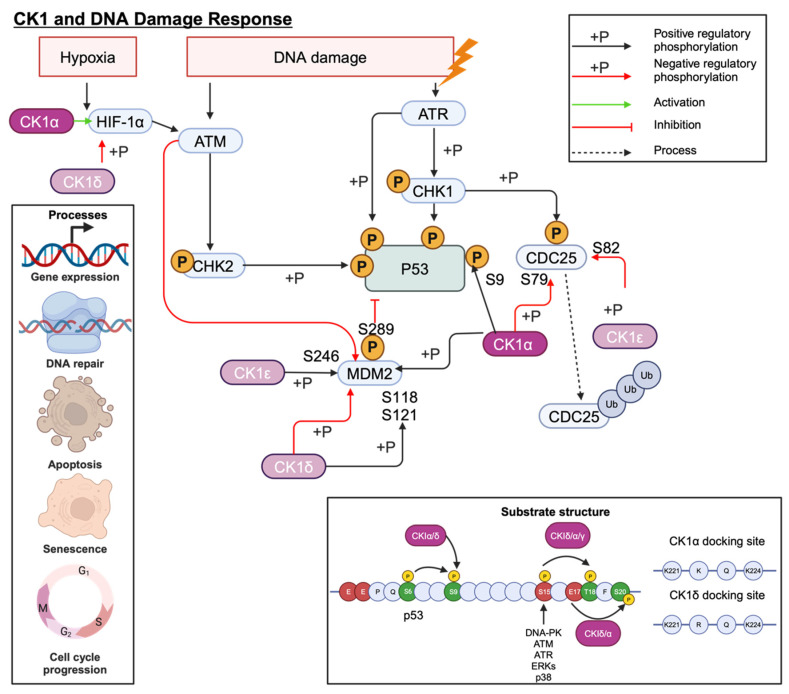
CK1 interplay with the P53 signalling cascade. CK1α phosphorylation is particularly important for regulating MDMX activity. Created with BioRender.com.

**Figure 11 cancers-16-02436-f011:**
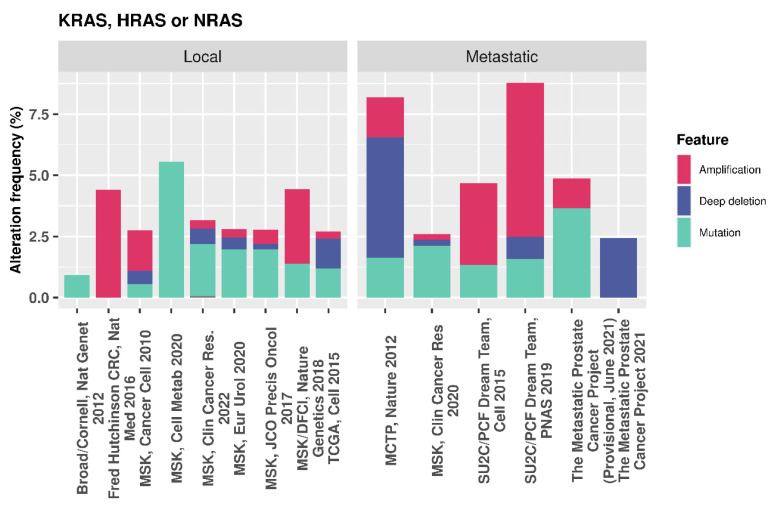
cBioPortal data showing the genetic alteration frequency in PCa for multiple RAS genes.

**Figure 12 cancers-16-02436-f012:**
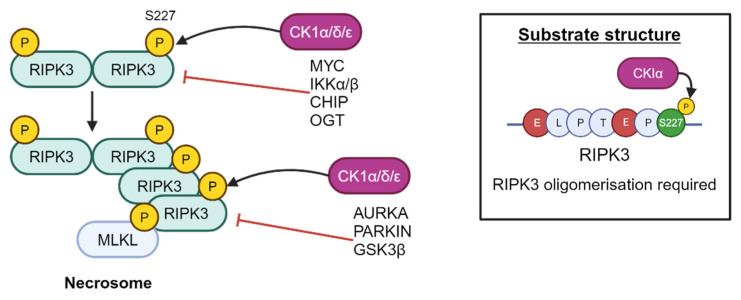
RIPK3 is a substrate of CK1α, which phosphorylates RIPK3 at residue S227. Created with BioRender.com.

**Figure 13 cancers-16-02436-f013:**
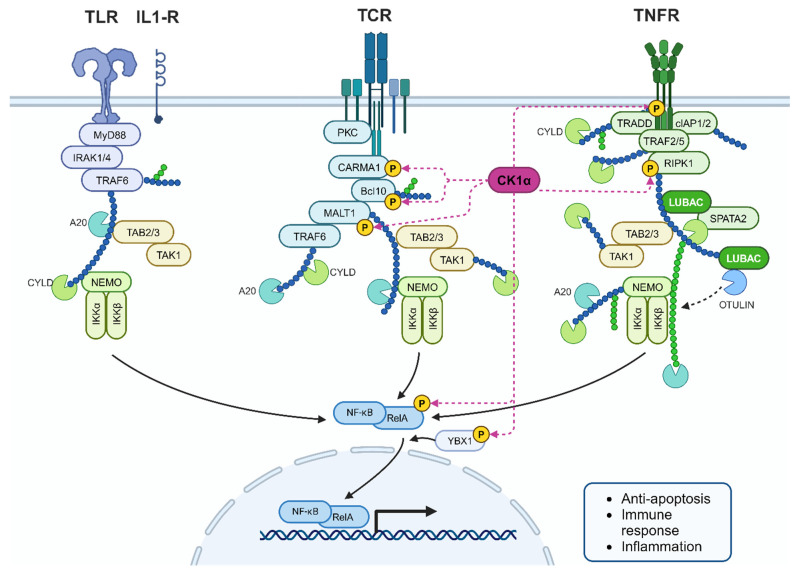
CK1α interplay in the NF-κB pathway. CK1α phosphorylates many cellular components within the NF-κB signalling cascade to regulate the pathway. RIPK1 is also a substrate of CK1α as part of Tumour Necrosis Factor Receptor (TNFR) signalling. De-regulated CK1α could contribute to increased inflammation in PCa. Created with BioRender.com.

**Table 1 cancers-16-02436-t001:** Summary of the roles CK1 isoforms play in disease, including multiple cancer types, neurodegenerative diseases and metabolic disorders.

Disease	Role of CK1	Pathways of Interest	Isoforms	References
**Prostate cancer**	Upregulated Contributes to enzalutamide resistance via ATM	AR signalling Wnt signalling Hippo signaling P53 signalling NF-kB signalling	*CSNK1A1*	[39]
**Breast cancer**	Upregulated, isoform varies by subtype Promotes metastasis Contributes towards 4-Hydroxytamoxifen sensitivity	ER signalling Wnt signalling PI3K/AKT/mTOR pathway	*CSNK1A1* *CSNK1D* *CSNK1G2*	[40,41]
**Myelodysplastic syndrome**	Mutated Heterozygous 5q deletion Loss contributes to lenalidomide sensitivity	Wnt signalling	*CSNK1A1*	[37]
**Colorectal cancer**	Mutation Amplification	Wnt signalling P53 signalling	*CSNK1A1* *CSNK1D* *CSNK1E*	[6]
**Other cancers: ovarian, pancreatic, liver, lung, head and neck, renal, stomach, cervical**	Mutation or changes in expression promote tumourigenesis via altered signalling		*CSNK1A1* *CSNK1D* *CSNK1E* *CSNK1G2*	[8]
**Alzheimer’s disease**	Upregulated	Amyloid plaque formation Tau hyperphosphorylation	*CSNK1A1* *CSNK1D* *CSNK1E*	[6,8,42]
**Parkinson’s disease**	Upregulated	Dopamine signalling cAMP-regulated neuronal phosphoprotein 32 cascade	*CSNK1A1*	[6,8,42]
**Metabolic disorders**	Upregulated	mTOR pathway	*CSNK1D* *CSNK1E*	[8]

**Table 2 cancers-16-02436-t002:** Human Protein Atlas consensus prostate tissue transcriptomic data for each of the CK1 isoforms [22]. Expression of *CSNK1A1* is highest of all the isoforms in normal prostate tissue, with expression of the γ isoforms at a very low level. Differential expression could pose an opportunity for targeting of specific isoforms based upon abundance.

Isoform	Normalised Transcripts per Million (nTPM)
*CSNK1A1*	85.4
*CSNK1D*	50.2
*CSNK1E*	77.8
*CSNK1G1*	3.6
*CSNK1G2*	40
*CSNK1G3*	10.7

**Table 3 cancers-16-02436-t003:** CK1-mediated predicted phosphorylation sites on the androgen receptor. Position and code identify the amino acid residue phosphorylated on AR as well as the underline within the peptide sequence. The score refers to the strength of the prediction, with higher scores indicating increased potential for phosphorylation of that residue. No experimental evidence is available in the Group-Based Prediction System V6.0 for CK1 phosphorylation of AR.

Position	Code	Kinase	Substrate	Peptide Sequence	Score
219	T	CK1	AR	REASGAPTSSKDNYL	0.3454
650	T	CK1	AR	EGEASSTTSPTEETT	0.314
516	S	CK1	AR	VSRVPYPSPTCVKSE	0.2799
530	S	CK1	AR	EMGPWMDSYSGPYGD	0.2483
651	S	CK1	AR	GEASSTTSPTEETTQ	0.1831
498	T	CK1	AR	AGQESDFTAPDVWYP	0.182
82	T	CK1	AR	QQQQQQETSPRQQQQ	0.1517
495	S	CK1	AR	QGLAGQESDFTAPDV	0.1082
542	T	CK1	AR	YGDMRLETARDHVLP	0.0984
568	S	CK1	AR	LICGDEASGCHYGAL	0.0931
518	T	CK1	AR	RVPYPSPTCVKSEMG	0.0909
783	S	CK1	AR	MHKSRMYSQCVRMRH	0.0687
306	T	CK1	AR	AGKSTEDTAEYSPFK	0.0683
397	S	CK1	AR	ENPLDYGSAWAAAAA	0.068
233	S	CK1	AR	LGGTSTISDNAKELC	0.0673
178	S	CK1	AR	FPGLSSCSADLKDIL	0.0648
303	T	CK1	AR	DDSAGKSTEDTAEYS	0.063
229	T	CK1	AR	KDNYLGGTSTISDNA	0.0611
760	S	CK1	AR	RSFTNVNSRMLYFAP	0.0542

**Table 4 cancers-16-02436-t004:** A subset of available CK1 small-molecule inhibitors. Many inhibitors are non-specific or are dual-target inhibitors for CK1δ/ε.

Inhibitor	CK1 Isoforms Targeted	Additional Targets	References
CKI-7	Non-specific	PRK2, AMPK, CHK1, PLK1	[87,104]
D4476	Non-specific	ALK5	[87]
IC261	CK1δ/ε	CDK5, microtubules	[105]
PF670462	CK1ε	EGFR	[106]
BTX-A51 (phase I)	CK1α	CDK7, CDK9	[107]

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
