# Peer review of "Casein Kinase 1α—A Target for Prostate Cancer Therapy?"

_cancers, 2024, doi:10.3390/cancers16132436_

Round 1

Reviewer 1 Report

Comments and Suggestions for Authors

The manuscript by Walkar et al. provides a comprehensive review of the deregulation of CK1 in cancer, its effects on signalling pathways, and its role in prostate tumorigenesis. The authors specifically highlight the CK1α isoform as a potential therapeutic target for PCa. The manuscript is very nicely written.  It provides detailed information that will be very valuable for PCa research.

I have a few comments for polishing the manuscript.

Major Comments:

  1. The authors should add a detailed paragraph on prostate cancer in the introduction section.
  2. The authors should add some information on CRPC in the Introduction section.
  3. There are some text errors throughout the manuscript, e.g., Page 2, Lines 63, 72, Page 3, Line 96 etc. The authors should correct these errors.
  4. The authors can add a table to highlight the role of CK1 in various diseases.

Author Response

Comment 1: The authors should add a detailed paragraph on prostate cancer in the introduction section.

Comment 2: The authors should add some information on CRPC in the Introduction section.

We were unsure as to the level of detail required by reviewer 1 with respect to both prostate cancer and castration resistant prostate cancer. To try and address these two points we have amended the first paragraph and added in some extra details highlighted in yellow. If this is not what the reviewer had in mind could they please offer some more guidance as to what extra detail they think should be included.

Changes can be found from lines 34-50 on page 1 and 2.

Prostate cancer (PCa) results in the deaths of more than 12,000 men each year, with as many as 1 in 6 men diagnosed throughout their lifetime. Indeed, incidence rates are expected to rise, increasing by 15% in 2040 due to an aging population and improved detection methods [1]. A key driver of PCa is androgen receptor (AR) signalling (Figure 1) therefore, therapies targeting this pathway are central to the management of advanced PCa [2]. In particular, androgen deprivation therapy (ADT) is used to reduce the levels of the AR ligand, testosterone, in the body to switch off the AR signalling pathway. Whilst this may be initially effective, resistance to these treatments can arise through multiple mechanisms including AR amplification or mutation, AR splice variant production and alterations to AR co-regulators demonstrating that AR is still a tractable therapeutic target in therapy resistant disease. Patients that are no longer responsive to ADT are described as having castration resistant PCa (CRPC) [3]. Unfortunately, for the 10-20% of patients which progress to CRPC the outlook is poor despite the development of second-generation anti-androgen treatments, such as enzalutamide, which target the AR directly to prevent aberrant signalling. This highlights the unmet clinical need for novel therapies to be used alone or in combination with existing ADT. To achieve this, additional key drivers which are potentially targetable need to be identified and characterized within the context of PCa.

Comment 3. There are some text errors throughout the manuscript, e.g., Page 2, Lines 63, 72, Page 3, Line 96 etc. The authors should correct these errors.

All of these formatting errors that have arisen during conversion of the original document have now been corrected throughout the manuscript. Specific location of corrections are listed below:

Line 94 – alpha symbol re-inserted

Line 150 – hieroglyphic removed

Line 210 – Typo corrected: “serine” converted to “Serine”

Line 300 – gamma symbols re-inserted

Line 306 – hieroglyphic removed; alpha inserted

Line 456, 634, 635,637,638 – RB1 changed to Rb1 to be consistent throughout manuscript

Line 464 – moved Figure 10 citation before reference 82 citation.

Line 562, 565, 566, 567, 577,578, 581, 583, 584, 586, 588 – replaced hieroglyph with a Kappa symbol

Line 643 – removed hieroglyph and inserted a Beta symbol

Comment 4. The authors can add a table to highlight the role of CK1 in various diseases.

Table 1 has been inserted to include a summary of the roles of CK1 in disease. This can be found on page 8.

Reviewer 2 Report

Comments and Suggestions for Authors

The review article of Emma Lishman-Walker and Kelly Coffeyis summarised current literature data on research iof the functional role and therapeutic potential of protein kinase CK1α for the prostate cancer treatment. The topic of the article is highly relevant, given the widespread prevalence of prostate cancer and the low sensitivity of this type of oncological pathology to classical treatment regimens in late stages. This requires a search for new targets for the treatment of prostate cancer diseases, and protein kinase CK1α can be considered as a promising new target due to impact on signalling pathways and how this contributes towards 28 prostate tumourigenesis. This review is well illustrated, the material is logically grouped and it can certainly be recommended for publication after taking into account the following comments and suggestions.

1. It would be useful to illustrate the article with the specific examples of known protein kinase CK1α substrates, showing the chemically important functional groups for binding.

2. Lines 65, 72, 96 and further - unreadable hieroglyphic error notes before literature references should be removed.

Author Response

Comment 1: It would be useful to illustrate the article with the specific examples of known protein kinase CK1α substrates, showing the chemically important functional groups for binding.

Specific examples have now been included to show the chemically important groups of residues within the protein substrates. These have been embedded within the specific signalling sections to which they relate. Accompanying text has been included within each section to justify their inclusion. 

Line 206-208 – “The impact of substrate availability on kinase activity became evident when modelled in E.coli, as CK1 auto-phosphorylation rate varied dependent upon the levels of available substrate [25].”moved from further down the paragraph to accommodate information regarding consensus binding sites.

Line 209 – inserted “canonical” to support binging site information.

Line 211 – “non-canonical” inserted to support binding site information.

Line 344-348: Inserted “β-catenin is an example of a non-phosphorylation primed CK1α substrate. Instead, it contains an acidic cluster 5-11 residues downstream of S45 and a hydrophobic side chain in the +1 position. Furthermore, Y145 within the first Armadillo domain of the protein is important for successful phosphorylation at S45 as a consequence of α-catenin binding regulation”

Lines 351-354: Inserted “Four residues, namely S51, S54, S57 and T60, were identified as critical for axin phos-phorylation by CK1α which match the consensus CK1 phosphorylation sequence with T60 being critical for attenuation of axin-tankyrase binding (Figure 8)”

Line 358 – Substrate diagram added to Figure 8

Line 471-473 – Inserted “Indeed, p53 can also be phosphorylated by multiple CK1 isoforms in primed phosphorylation and non-primed phosphorylation events (Figure 10)”

Line 504 – added substrate diagram to Figure 10

Line 553-554 – Inserted “This phosphorylation is dependent on E221 and E225 and RIPK3 oligomerisation (Figure 12)”

Line 558 – RIPK3 substrate diagram inserted as Figure 12

Comment 2: Lines 65, 72, 96 and further - unreadable hieroglyphic error notes before literature references should be removed.

All of these formatting errors that have arisen during conversion of the original document have now been correct throughout the manuscript. Specific location or corrections are listed below 

Line 94 – alpha symbol re-inserted

Line 150 – hieroglyphic removed

Line 210 – Typo corrected: “serine” converted to “Serine”

Line 300 – gamma symbols re-inserted

Line 306 – hieroglyphic removed; alpha inserted

Line 456, 634, 635,637,638 – RB1 changed to Rb1 to be consistent throughout manuscript

Line 464 – moved Figure 10 citation before reference 82 citation.

Line 562, 565, 566, 567, 577,578, 581, 583, 584, 586, 588 – replaced hieroglyph with a Kappa symbol

Line 643 – removed hieroglyph and inserted a Beta symbol

Round 2

Reviewer 1 Report

Comments and Suggestions for Authors

The manuscript by Walkar et al. provides a comprehensive review of the deregulation of CK1 in cancer, its effects on signalling pathways, and its role in prostate tumorigenesis. The authors specifically highlight the CK1α isoform as a potential therapeutic target for PCa. The manuscript is very nicely written.  It provides detailed information that will be very valuable for PCa research.

The authors have addressed all the previous comments. Thus, the manuscript can be accepted in its present form.